# Phytoplankton responses to changing temperature and nutrient availability are consistent across the tropical and subtropical Atlantic

Cristina Fernández-González [1,2], Glen A. Tarran[3], Nina Schuback [4], E. Malcolm S. Woodward [3], Javier Arístegui[5] & Emilio Marañón [1,2 ✉]

Temperature and nutrient supply interactively control phytoplankton growth and productivity, yet the role of these drivers together still has not been determined experimentally over large spatial scales in the oligotrophic ocean. We conducted four microcosm experiments in the tropical and subtropical Atlantic (29°N-27°S) in which surface plankton assemblages were exposed to all combinations of three temperatures (in situ, 3 °C warming and 3 °C cooling) and two nutrient treatments (unamended and enrichment with nitrogen and phosphorus). We found that chlorophyll *a* concentration and the biomass of picophytoplankton consistently increase in response to nutrient addition, whereas changes in temperature have a smaller and more variable effect. Nutrient enrichment leads to increased picoeukaryote abundance, depressed *Prochlorococcus* abundance, and increased contribution of small nanophytoplankton to total biomass. Warming and nutrient addition synergistically stimulate light-harvesting capacity, and accordingly the largest biomass response is observed in the warmed, nutrient-enriched treatment at the warmest and least oligotrophic location (12.7°N). While moderate nutrient increases have a much larger impact than varying temperature upon the growth and community structure of tropical phytoplankton, ocean warming may increase their ability to exploit events of enhanced nutrient availability.

[1] Departamento de Ecoloxía e Bioloxía Animal, Universidade de Vigo, Vigo, Spain. [2] Centro de Investigacións Mariñas, Universidade de Vigo, Vigo, Spain. [3] Plymouth Marine Laboratory, Plymouth, UK. [4] Swiss Polar Institute, Sion, Switzerland. [5] Instituto de Oceanografía y Cambio Global, Universidad de Las Palmas de Gran Canaria, Las Palmas de Gran Canaria, Spain. ✉email: em@uvigo.es

Oligotrophic tropical and subtropical regions cover more than 60% of the global ocean and contribute significantly to marine primary production[1,2] and downward carbon export[3,4]. These regions are characterized by a strong, persistent thermal stratification associated with a reduced nutrient supply into the euphotic layer, which leads to low phytoplankton standing stocks dominated by small cells (picophytoplankton) forming the base of a complex microbial food web[5]. Satellite data suggest that, due to ongoing climate warming, oligotrophic regions of all major ocean basins are expanding[6,7] and displaying decreasing trends in phytoplankton biomass[8]. Although most biogeochemical models predict reduced phytoplankton biomass and productivity as a result of increased stratification in low-latitude oligotrophic regions[9,10], there are also simulations that suggest increasing net primary production[11] and even larger picophytoplankton standing stocks[12] under future, ocean warming scenarios. The expected response of phytoplankton growth and metabolism to changing hydroclimatic conditions in the tropical and subtropical ocean thus remains highly uncertain[13,14].

Temperature and nutrient supply are key environmental drivers that control phytoplankton growth and productivity[15–17]. In addition to its impact on vertical mixing and nutrient supply into the euphotic zone, temperature also influences phytoplankton growth directly through its effect on the kinetics of metabolic reactions[18]. The effects of temperature and nutrient supply on biological rates are often interactive[19] and there is growing evidence that the temperature sensitivity of phytoplankton growth and metabolism is reduced when nutrient availability is low. Laboratory experiments with cultures of cyanobacteria, hapto-phytes and diatoms have shown that the effect of temperature on metabolic rates and nutrient stoichiometry is weaker under nutrient limitation than under nutrient-replete conditions[20–22]. Comparatively less is known about the interactive effect of temperature and nutrient availability in natural phytoplankton assemblages. O'Connor et al.[23], in a microcosm experiment with a coastal plankton community, demonstrated that warming stimulates phytoplankton primary productivity after nutrient enrichment but not under in situ nutrient-depleted conditions. Similarly, Liu et al.[24], in a series of temperature-modulated dilution experiments in the East China Sea, found that the temperature sensitivity of Synechococcus growth rates under in situ oligotrophic conditions was lower than under nutrient-replete conditions. A recent modelling and observational study on the effects of heatwaves on phytoplankton dynamics in temperate and tropical regions concluded that phytoplankton blooms during events of increased seawater temperature are weaker when background nutrient concentrations are low and stronger when nutrient concentrations are high[25]. The biogeographical patterns of phytoplankton nutrient limitation in the Atlantic Ocean are well delineated, thanks to both multiple-nutrient bioassays[26–28] and metagenomic analyses[29]. However, despite the expected significance of temperature–nutrient interactive effects on phytoplankton growth and metabolism in oligotrophic regions, there has yet been no systematic, experimental assessment of the combined role of these two drivers over large spatial scales in the tropical and subtropical Atlantic.

Picophytoplankton contribute >50% of the phytoplankton biomass in the tropical and subtropical Atlantic Ocean[30–32]. While sometimes considered as a single functional group, pico-phytoplankton are phylogenetically and functionally diverse, including the cyanobacteria Prochlorococcus and Synechococcus and the picoeukaryotes, all of which possess distinct ecophysio-logical traits[30,33,34]. Recent evidence based on both laboratory data and field observations indicates that these three picophyto-plankton components differ in their thermal response traits[35,36], growth rates[37,38], and nutrient uptake strategies[39,40]. The contrasting responses of Prochlorococcus, Synechococcus and picoeukaryotes to variations in nutrient supply have been investigated[26,28,31,41,42] but the combined effect of changing temperature and nutrient availability on the growth and biomass of these groups across the tropical Atlantic Ocean is unknown.

A critical issue regarding the fate of tropical phytoplankton in a future warmer ocean is that species living in low-latitude regions typically display optimal growth temperatures that are close to the current mean temperatures they are already experiencing[43]. This means that, because thermal response curves show strong negative skewness (growth rates decline faster above the optimum temperature than below), the anticipated warming of the ocean may have particularly negative impacts on the performance of tropical species. It can then be hypothesized that phytoplankton assemblages in the warmest ocean regions will be the ones suffering most from increased temperature, or the ones that benefit least from enhanced nutrient availability under warming conditions.

Climate change and its impacts on organisms and ecosystems manifest themselves not only as sustained, multidecadal trends[44] but also through the increasing frequency and intensity of extreme events[45,46]. Marine heatwaves and tropical storms (cyclones and typhoons) cause rapid changes in sea surface temperature and vertical mixing that can also affect the availability of both nutrients and irradiance for phytoplankton[25,47,48]. Short-term factorial experiments can thus provide mechanistic insight into the potential impacts of extreme events on phytoplankton growth, biomass and community structure[49].

To determine the concurrent effects of changes in temperature and nutrient availability on tropical and subtropical phytoplankton communities, we conducted four microcosm experiments across the central Atlantic Ocean (29°N–27°S) in which surface plankton assemblages were exposed to all combinations of three temperatures (in situ, 3 °C cooling and 3 °C warming) and two nutrient treatments (unamended and addition of nitrogen and phosphorus). Our main goals were: (i) to ascertain the relative role of temperature and nutrient supply as drivers of phytoplankton biomass and growth, (ii) to test the hypothesis that assemblages from warmer regions are most vulnerable to increased temperatures and (iii) to examine group-specific differences in the response of picophytoplankton to changes in temperature and nutrient availability.

## Results

**Initial conditions in situ.** All experiment sites displayed warm and strongly stratified conditions, with sea surface temperatures >21 °C and low surface nutrient concentrations (nitrate + nitrite ≤ 0.05 μmol L$^{-1}$), as well as low surface chlorophyll $a$ concentrations (≤0.25 μg L$^{-1}$) characteristic of oligotrophic waters (Table 1 and Fig. 1). However, the degree of oligotrophy varied markedly among sites as the cruise track crossed the Equatorial upwelling region (15°N–5°S), which is characterized by a shallower thermocline and nutricline that are associated also with a shallower deep chlorophyll maximum (DCM) (Fig. 1b). Thus, the station at 12.7°N was significantly less oligotrophic than the other sites, as indicated by its high value of the resource supply index (RSI) as well as its shallow 1%PAR depth (Table 1), which suggests a relatively large phytoplankton standing stock. This was confirmed by the values of surface Chl $a$ and pico-phytoplankton biomass, which were highest at 12.7°N (0.25 μg L$^{-1}$ and 26 μg C L$^{-1}$, respectively). The site at 7.3°S also showed the influence of the equatorial upwelling, albeit less intensely than found at 12.7°N, as reflected in its intermediate values of nitracline depth, DCM depth, and RSI. Conversely, the stations at 28.7°N and 26.7°S were the most oligotrophic, showing

**Table 1 Initial conditions at the experimental sites.**

|  | Exp. 1 | Exp. 2 | Exp. 3 | Exp. 4 |
|---|---|---|---|---|
| Latitude | 28.7°N | 12.7°N | 7.3°S | 26.7°S |
| Longitude | 33°W | 28.5°W | 25°W | 25.8°W |
| Temperature (°C) | 25.5 | 28.4 | 25.8 | 21.5 |
| Salinity | 37.5 | 36.0 | 36.4 | 36.3 |
| $[NO_3^- + NO_2^-]$ ($\mu$mol L$^{-1}$) | <0.02 | 0.05 | <0.02 | <0.02 |
| $[HPO_4^{2-}]$ ($\mu$mol L$^{-1}$) | <0.02 | <0.02 | 1.13 | 1.29 |
| 1% PAR$_z$ (m) | 142 | 56 | 97 | 118 |
| Nitracline$_z$ (m) | 146 | 15 | 94 | 128 |
| DCM$_z$ | 115 | 43 | 86 | 120 |
| $[NO_3^- + NO_2^-]$ at 1%PAR$_z$ ($\mu$mol L$^{-1}$) | 0.5 | 24.7 | 1.3 | 0.2 |
| RSI (mmol N kg$^{-1}$) | 0.8 | 15.4 | 2.0 | 0.6 |
| Chl $a$ ($\mu$g L$^{-1}$) | 0.09 (0.00) | 0.25 (0.05) | 0.13 (0.01) | 0.09 (0.01) |
| $F_v/F_m$ | 0.34 (0.01) | 0.35 (0.04) | 0.34 (0.03) | 0.4 (0.02) |
| Total phytoplankton C ($\mu$g C L$^{-1}$) | 12.6 (0.3) | 34.5 (7.3) | 18.1 (1.2) | 12.9 (0.7) |
| Large nano + microphyto C ($\mu$g C L$^{-1}$) | 0.14 (0.08) | 0.89 (0.50) | 0.68 (0.07) | 0.27 (0.02) |
| Small nanophyto C ($\mu$g C L$^{-1}$) | 7.9 (0.5) | 8.0 (8.5) | 7.0 (1.6) | 3.2 (1.0) |
| Picophytoplankton C ($\mu$g C L$^{-1}$) | 4.5 (0.1) | 25.6 (0.6) | 10.5 (0.3) | 9.4 (0.3) |
| *Prochlorococcus* C contribution (%) | 67.1 (1.5) | 66.6 (2.8) | 74.3 (2) | 66.6 (2.1) |
| *Synechococcus* C contribution (%) | 3.7 (0.2) | 13.5 (0.7) | 1.6 (0.1) | 3.6 (0.1) |
| Picoeukaryote C contribution (%) | 29.3 (2.2) | 19.9 (0.6) | 24.1 (2.6) | 29.8 (3.2) |

Physico-chemical and biological data corresponding to the surface (2–5 m) of the four locations where experiments were conducted. Mean value and standard deviation (SD) are given for variables with replicate measurements. The resource supply index (RSI) was calculated as described in ref. [88] (see the "Methods" section). Total phytoplankton carbon biomass was estimated by assuming a C to Chl $a$ ratio (g:g) of 137[51] and the biomass of small nanophytoplankton (cells of 3–10 $\mu$m in ESD) was estimated as the difference between total phytoplankton carbon and the carbon of picophytoplankton (cells < 3 $\mu$m) plus large nano- and microphytoplankton (cells > 10 $\mu$m). Also shown is the contribution of *Prochlorococcus*, *Synechococcus* and picoeukaryotes to total picophytoplankton biomass.

a deep nitracline, 1%PAR level and DCM as well as a low RSI value. Accordingly, the station at 28.7°N had the lowest pico-phytoplankton biomass (4.5 $\mu$g C L$^{-1}$). We estimated (based on the measured Chl $a$ concentration) a surface phytoplankton carbon of 13–35 $\mu$g C L$^{-1}$ for the four locations, with picophy-toplankton and small nanophytoplankton contributing on average 58% and 36%, respectively, of total phytoplankton biomass (Table 1). Larger cells (>10 $\mu$m) represented a minor fraction (<8%) of total phytoplankton biomass in all stations. *Pro-chlorococcus* was by far the largest contributor to total picophy-toplankton biomass (65–75%), followed by the picoeukaryotes (20–30%) and *Synechococcus* (4–13%).

**Evolution of phytoplankton standing stocks**. To remove the effect of differences in initial phytoplankton abundance among the different sampling locations, we standardized Chl $a$ con-centration and picophytoplankton carbon biomass by dividing the values for each treatment, at each sampling time, by the initial value (i.e. at $t = 0$ h). The dynamics of standardized Chl $a$ and picophytoplankton biomass were relatively similar in all four experiments (Supplementary Figs. 1 and 2) and therefore their mean temporal evolution could be described by averaging all of the data for each treatment and sampling time (Fig. 2).

There was an initial decrease in phytoplankton stocks (24–48 h), more marked for picophytoplankton carbon than for Chl $a$, after which a clear increase took place in all treatments with added nutrients (Fig. 2). The initial decrease in picophyto-plankton carbon was particularly strong in the experiment at 28.7°N, and in the experiment at 12.7°N the warmed treatments displayed a more marked initial decrease than the other treatments (Supplementary Fig. 2). The response to the added nutrients in terms of Chl $a$ occurred earlier (48–72 h) than that observed in terms of picophytoplankton C (72–96 h). By the end of the experiment, the nutrient-amended treatments displayed a significantly larger concentration of Chl $a$ (by a factor of 2–2.5) and picophytoplankton biomass (ca. 50% more) than the non-amended treatments (Fig. 3). There was strong evidence for a nutrient effect on final Chl $a$ and biomass in all four experiments

($t$-Student's tests, $p < 0.01$). In contrast, temperature showed little or no effect on the evolution of either Chl $a$ or picophytoplankton biomass. Incubation temperature had no effect on final Chl $a$ in the treatments without added nutrients, but under nutrient-enriched conditions, the increase in Chl $a$ was sometimes modulated by temperature. Thus, warming led to a larger nutrient-induced Chl $a$ increase (relative to that observed under in situ temperature and $-3$ °C) at 12.7°N and cooling led to a more modest Chl $a$ response at 26.7°S (Fig. 3).

We investigated the effect of the different temperature and nutrient conditions on picophytoplankton biomass by calculating the ratio between the biomass in each treatment and the biomass in the control (in situ temperature, no nutrient addition) at $t = 96$ h. This ratio revealed that nutrient addition tended to cause increased picophytoplankton biomass relative to the control, irrespective of temperature (Fig. 4). The largest net biomass increase in response to nutrient addition (by a factor of 2–3) was measured in the least oligotrophic sites (12.7°N and 7.3°S), where moderate to strong evidence of a nutrient effect was revealed by post-hoc Dunnett tests (Supplementary Table 1). More modest increases (< 30%) took place in the experiments conducted at the most oligotrophic stations (28.7°N and 26.7°S). The effect of warming and cooling was smaller than that of nutrient addition and did not display a consistent pattern. Thus, warming gave way to an enhanced (relative to the control) picophytoplankton biomass at 12.7°N and 26.7°S but a decrease at 28.7°N and 7.3°S, whereas cooling induced a biomass increase at 12.7°N and 7.3°S and a reduction at 28.7°N and 26.7°S. In no experiment did we find evidence for a temperature effect on picophytoplankton biomass (Dunnett's post-hoc tests, $p \geq 0.08$, Supplementary Table 1).

Given the marked impact that nutrient enrichment had on phytoplankton biomass, we also investigated its effect on the relative contribution of picophytoplankton and small nanophy-toplankton to total phytoplankton biomass at $t = 96$ h (Fig. 5). The final biomass of larger cells (> 10 $\mu$m in equivalent spherical diameter) ranged between 0.2 and 2.1 $\mu$g C L$^{-1}$ (Supplementary Table 2), always representing a minor fraction (< 8%) of total

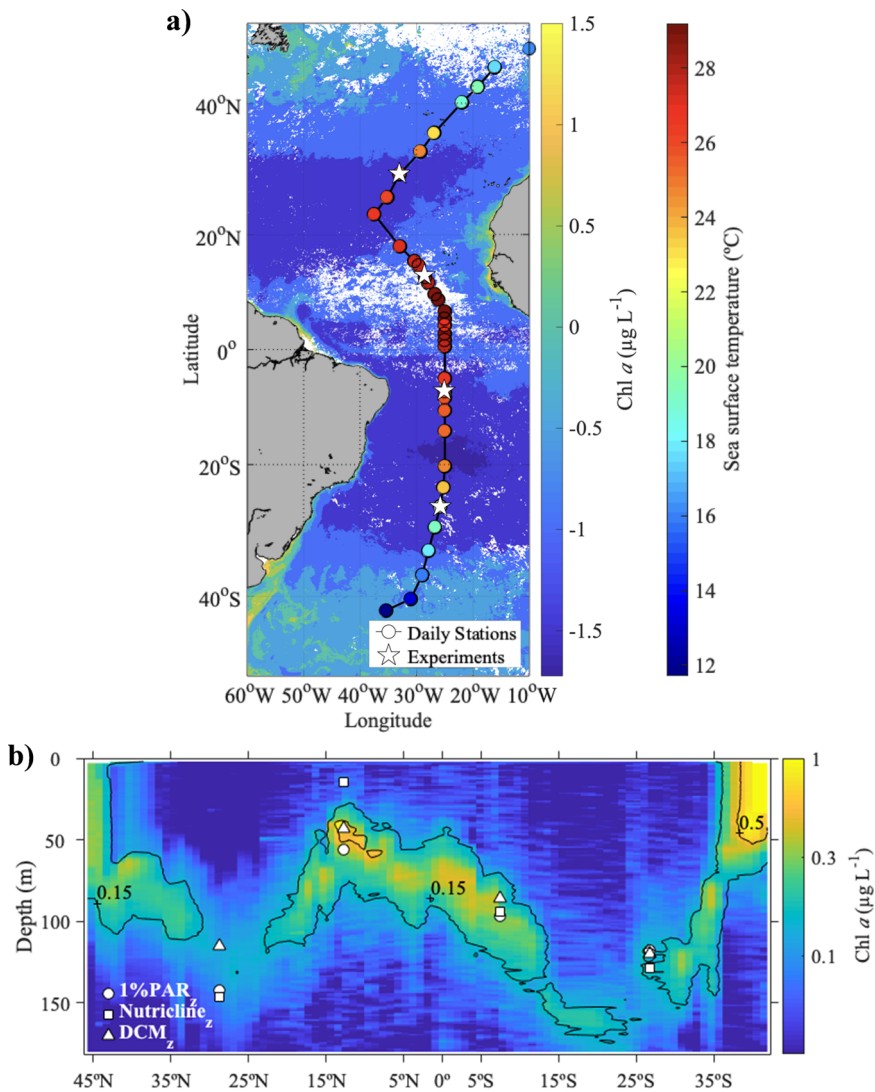

**Fig. 1 Oceanographic conditions along the Atlantic Meridional Transect 29. a** AMT 29 cruise track across the Atlantic Ocean. Circles mark the position of daily sampling stations, with their colour indicating sea surface temperature (see Supplementary Data 1). White stars correspond to the stations where samples for the experiments were taken, and the background colour map represents the surface mean chlorophyll *a* concentration from NASA Aqua-MODIS sensor (Ocean Biology Processing Group, Ocean Ecology Laboratory, NASA Goddard Space Flight Center, https://oceancolor.gsfc.nasa.gov/l3/order/) for the period October–November 2019. **b** Vertical distribution of fluorescence-derived chlorophyll *a* concentration ($\mu$g L$^{-1}$) along the transect (see Supplementary Data 2). Different symbols indicate the depth of the 1% PAR level (circles), the nitracline (squares), and the deep chlorophyll maximum (triangles) at the locations where samples for the experiments were collected.

biomass. In all experiments, irrespective of the C:Chl *a* ratio used to calculate total phytoplankton C, the biomass contribution by small nanophytoplankton increased after nutrient addition (Fig. 5). This increase in the contribution of 3–10 μm cells (reaching 80%) was particularly marked in the experiments that showed the largest response to the added nutrients (12.7°N and 7.3°S).

**Variability in photosystem II photochemical efficiency**. $F_v/F_m$ showed only small variations during the experiments and did not display a consistent pattern of response to either temperature or nutrient availability (Supplementary Fig. 3). After an early decrease during the first 24–48 h (particularly marked in the warmed treatments at 28.7°N and 12.7°N), $F_v/F_m$ tended to recover and, in most experiments, reached slightly higher values than the initial ones by 72 h. There was typically an overlap between the $F_v/F_m$ values measured in nutrient-enriched and non-enriched

treatments, with the notable exception of the experiment at 26.7°S (Supplementary Fig. 3d). At this site, *t*-tests revealed strong evidence for a nutrient-induced increase in photosystem II photochemical efficiency both at $t = 72$ h ($t(4) = -4.857$, $p = 0.008$) and $t = 96$ h ($t(16) = -4.73$, $p < 0.001$).

**Group-specific picophytoplankton responses**. The three groups of picophytoplankton investigated (*Prochlorococcus*, *Synechococcus* and picoeukaryotes) showed distinct patterns of response to the experimental treatments. The picoeukaryotes were most responsive to enhanced nutrient availability, as they increased their biomass (relative to the control) in the nutrient-enriched treatments of all four experiments, irrespective of temperature (Fig. 6). In 8 out of 12 instances (4 experiments, 3 nutrient-amended treatments), we found moderate to very strong evidence for a higher picoeukaryote biomass after nutrient addition than in the control (Dunnett post-hoc test, $p < 0.05$, Supplementary

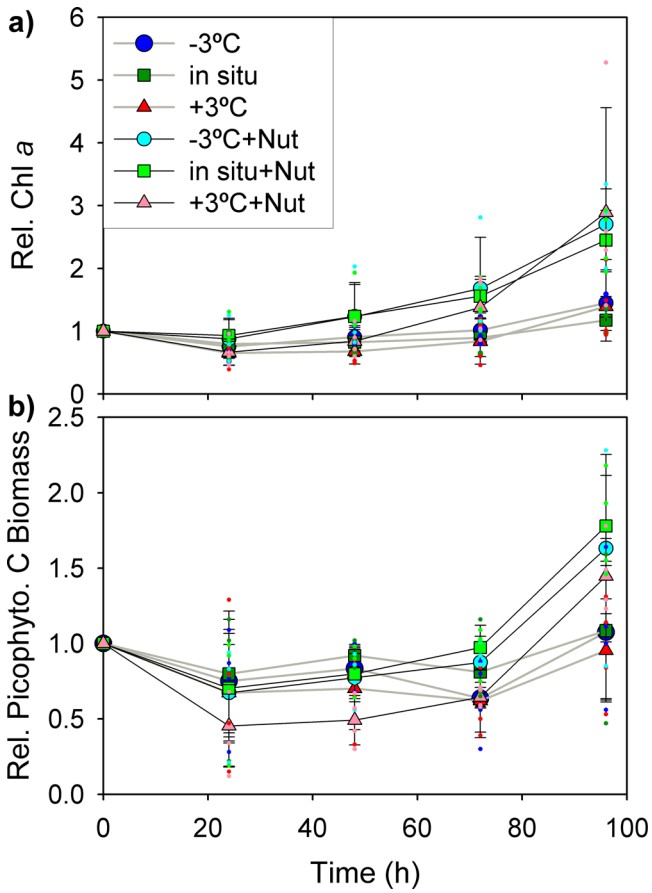

**Fig. 2 Mean evolution of chlorophyll *a* concentration and picophytoplankton carbon biomass.** Mean temporal evolution of relative (i.e. standardized by the initial value at each location) of **a** fluorescence-derived chlorophyll *a* concentration (Chl *a*) and **b** picophytoplankton carbon biomass during the four experiments. Symbols represent the 6 different treatments, with black and grey lines corresponding to treatments with (+Nut) and without nutrient addition, respectively. The three temperature treatments were: a decrease of 3 °C relative to in situ temperature (−3 °C), unchanged temperature (in situ), and an increase of 3 °C relative to in situ temperature (+3 °C). Data shown are the mean of the values obtained at each sampling time from all experiments and error bars represent the standard deviation ($n = 4$). Dots indicate the individual measurements obtained from each experiment.

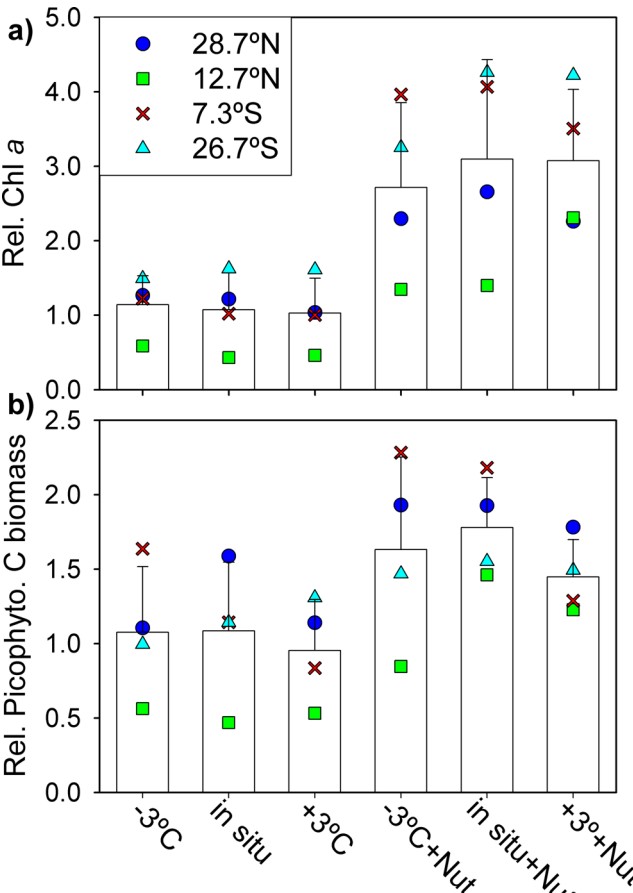

**Fig. 3 Effect of treatment on final chlorophyll *a* concentration and picophytoplankton carbon biomass. a** Relative Chl *a* concentration for each treatment at the end of the experiments ($t = 96$ h). Extracted Chl *a* concentration was divided by the initial value ($t = 0$ h) to remove differences among sampling locations. **b** Relative picophytoplankton C biomass. Symbols represent the mean value measured at the end of each individual experiment. Bars represent the mean value for all experiments together and error bars indicate the standard deviation ($n = 4$). *t*-test comparisons between nutrient-amended and non-amended treatments for both Chl *a* and picophytoplankton always yielded *p* values < 0.01.

Table 1). The largest nutrient-induced increase in picoeukaryote biomass (by a factor of 8 relative to the control) took place in conjunction with 3 °C warming at the warmest studied location (12.7°N) (Fig. 6f). At this site, there was strong evidence for a synergistic interaction between nutrient addition and warming on picoeukaryote biomass (Supplementary Table 3). Large picoeukaryote biomass increases (by a factor of >2) in response to nutrients were also measured under cooling and in situ temperature at 7.3°S (Fig. 6i). Changes in temperature alone did not cause consistent effects on picoeukaryotic biomass, and only in two cases (warming at 12.7°N and cooling at 7.3°S) did we find moderate evidence for a difference between treatment and control (Supplementary Table 1). In stark contrast to the behaviour of the picoeukaryotes, *Prochlorococcus* showed strong decreases (> 50%) in biomass under nutrient-enriched conditions at all temperatures (Fig. 6, Supplementary Table 1). Changes in temperature alone also affected *Prochlorococcus* biomass in several experiments, but the magnitude of the effect was smaller. The dominant response was that a change in temperature relative to in situ

conditions (either warming or cooling) tended to depress the abundance of *Prochlorococcus* relative to the control. Finally, *Synechococcus* displayed complex responses to temperature and nutrient changes, as both nutrient enrichment and warming or cooling resulted, depending on the location, in increases or decreases in its abundance. The largest increases in *Synechococcus* biomass (by a factor of >2) occurred under nutrient-enriched conditions at 28.7°N and 12.7°N (Fig. 6b, e), and yet the strongest decrease was also recorded under nutrient-enriched conditions (12.7°N, Fig. 6e). As was the case with *Prochlorococcus*, temperature changes tended to negatively affect the abundance of *Synechococcus*, with the exception of the warming treatment at 28.7°N (Fig. 6b) and 26.7°S (Fig. 6k).

Cell-specific red fluorescence is a proxy for cellular Chl *a* content and thus reflects the variability of resource allocation into light-harvesting pigment–protein complexes. We observed a consistent pattern, in all three picophytoplankton groups, whereby cellular fluorescence increased in response to enhanced nutrient availability, and this response was further stimulated when nutrient addition was combined with warming (Fig. 7). In nearly all cases, we found strong evidence that combined warmed

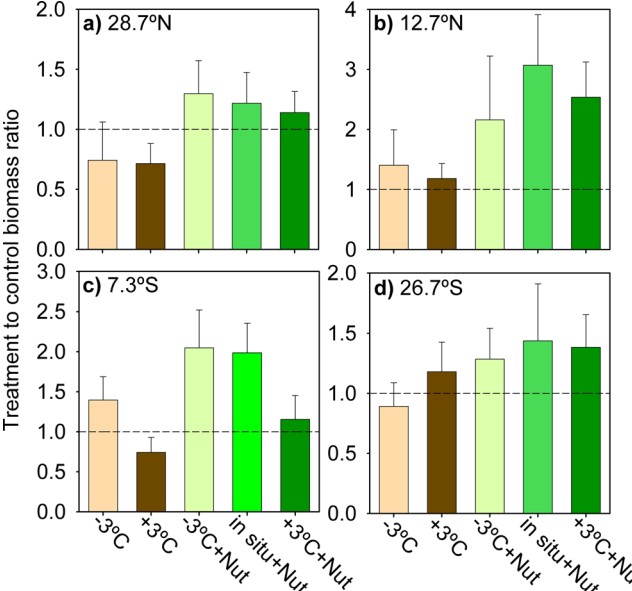

**Fig. 4 Picophytoplankton biomass response to each temperature and nutrient treatment.** Ratio between the picophytoplankton biomass in a given treatment divided by the biomass in the control (in situ temperature and no nutrient addition) at $t = 96$ h for the experiments conducted at **a** 28.7°N, **b** 12.7°N, **c** 7.3°S and **d** 26.7°S. Brown and green represent treatments without and with nutrient additions, respectively. Colour tones denote temperature treatments. Error bars show the standard deviation calculated by the propagation of uncertainty. The dashed line indicates a treatment-to-control ratio of 1. Note differences in the y-axis scale.

and nutrient-enriched conditions resulted in higher cellular fluorescence than observed in the control (Dunnett post-hoc test, $p < 0.01$, Supplementary Table 4). This response was strongest in the experiment conducted at 12.7°N, where for all picophytoplankton groups cellular fluorescence in the +3 °C +Nut treatment increased by a factor of > 3 relative to the control (Fig. 7, Supplementary Table 5). Temperature also caused consistent changes in cellular fluorescence, with cooling and warming predominantly inducing decreases and increases, respectively (Fig. 7). Considering the mean responses observed in all experiments together, nutrient addition together with warming increased cell fluorescence by a factor of > 2, while nutrient enrichment, alone or with cooling, and warming alone caused an increase of ca. 30% (Supplementary Table 5). We found strong evidence in all experiments for an interaction effect between temperature and nutrient availability on cellular fluorescence in one or several picophytoplankton groups (Supplementary Table 6). The dominant type of interaction between warming and nutrient addition was positive synergism, that is, the combined effect of the two drivers exceeded their additive effect (Fig. 7). The dominant type of interaction between cooling and nutrient addition was positive antagonism, which means that the observed response was less positive than predicted additively (Fig. 7).

## Discussion

Our experiments were conducted over a 6200-km transect spanning a wide range of oceanographic conditions and yet provided remarkably consistent patterns in the short-term response of phytoplankton to combined changes in temperature and nutrient supply: (i) nutrient availability played a much larger role than temperature as a driver of phytoplankton growth dynamics; (ii) different picophytoplankton groups exhibited

contrasting responses to nutrient enrichment, with the picoeukaryotes being the most responsive group, and (iii) picophytoplankton cellular chlorophyll content was stimulated by both increased temperature and nutrient addition, an effect synergistically enhanced under combined warming and nutrient enrichment.

The observed stimulatory effect of nitrogen and phosphorus additions on phytoplankton abundance and biomass confirmed our expectations, considering the well-known nutrient-limited status of the subtropical and tropical Atlantic Ocean[27]. Given that nitrogen is the primary limiting nutrient for phytoplankton growth in the studied locations[26,27,50], the amount of nitrogen added in our experiments (2 $\mu$mol L$^{-1}$) could be expected to cause a Chl $a$ increase of at least 1.2 $\mu$g L$^{-1}$, assuming Redfield C:N stoichiometry and a C:Chl $a$ ratio of 137 g:g[51]. A larger Chl $a$ increase would be likely, given that C:Chl $a$ decreases with increased nutrient availability[52]. The comparatively modest Chl $a$ increase observed (ca. 3-fold on average, equivalent to a net increase of roughly 0.2–0.6 $\mu$g Chl $a$ L$^{-1}$) suggests a significant consumption of photoautotrophic biomass during the incubations, presumably due to grazing by protists, which is the largest loss process for phytoplankton in the tropical and subtropical Atlantic[53,54].

The observation that both total Chl $a$ and picophytoplankton abundance responded to the added nutrients only after 48–72 h has implications for the interpretation of short (24 h) nutrient-amended incubations such as those typically used in the dilution method to determine phytoplankton growth and microzooplankton grazing[55]. In the literature, the lack of difference in phytoplankton growth rates between nutrient-enriched and unamended incubations is often interpreted as implying the absence of nutrient limitation of growth. While this conclusion may be appropriate in more productive regions, our observations in the tropical Atlantic strongly suggest that phytoplankton inhabiting oligotrophic waters require time scales longer than 24 h to respond with a net biomass increase to enhanced nutrient availability. This delayed response may be due to both ecological and physiological constraints: the tight trophic coupling between phytoplankton growth and protist grazing in oligotrophic waters contributes to moderate the increase of photoautotroph biomass[56], low phytoplankton cell abundance makes it less likely that fast-growing taxa are present in sufficient numbers to cause a biomass increase[57,58], and heavily depleted internal nutrient quotas mean that a longer time is required for cells to upregulate their biosynthetic machinery[59].

Light-harvesting complexes and the catalysts involved in the light reactions of photosynthesis account for a large fraction of cellular nitrogen, and therefore cells regulate their content of chlorophyll $a$ depending on nitrogen availability[52]. In our experiments, all groups of picophytoplankton increased their chlorophyll fluorescence per cell under nutrient-enriched conditions, as has been found before in the tropical and subtropical Atlantic[26,28] and Pacific[58] oceans. This result supports the view that in the oligotrophic ocean nutrient scarcity leads not only to the limitation of phytoplankton standing stocks (Liebig limitation) but also to physiological impairment of photosynthetic cells, which process energy and resources at a slower rate than they are potentially capable of (Blackman limitation). The enhanced light-harvesting capacity in the nutrient-enriched treatments was not associated, in general, with increases in photosystem II quantum yield ($F_v/F_m$), which reflects the fact that under chronic macronutrient stress photochemical efficiency tends to be independent of nutrient-limited growth rate[60].

In addition to nutrient status, temperature also modulates resource allocation into light-harvesting complexes, because of the different temperature sensitivity of the light and dark

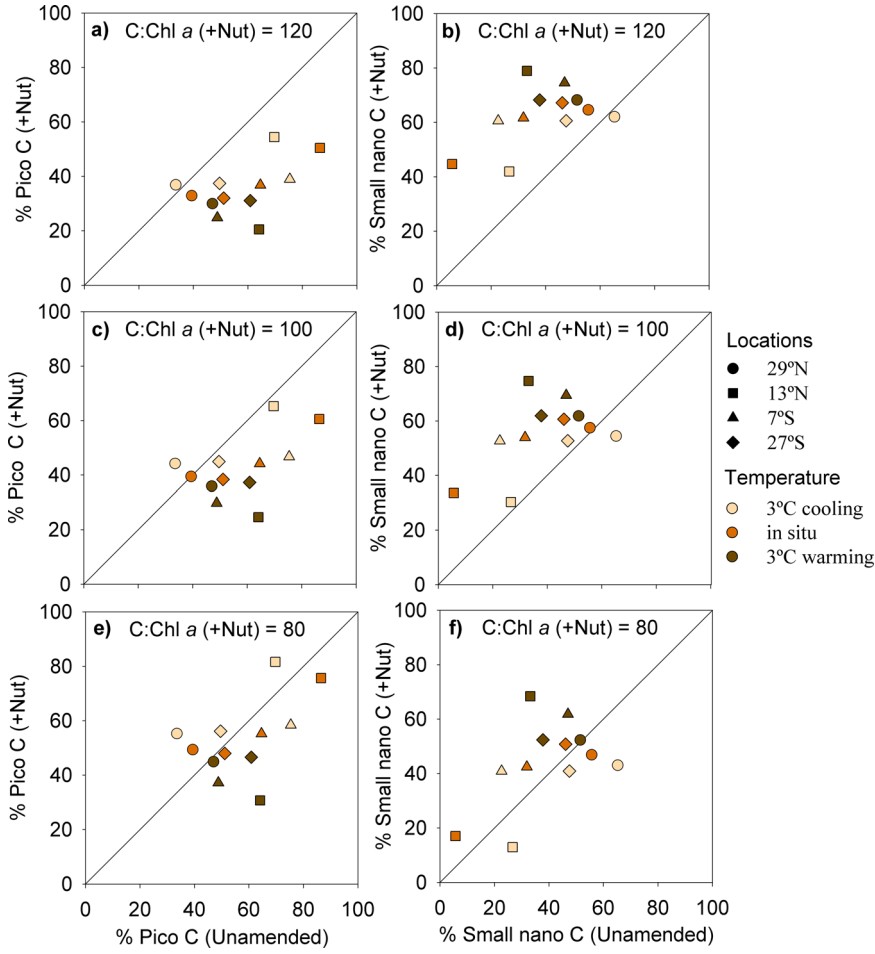

**Fig. 5 Effects of nutrient addition on the estimated biomass contribution of pico- and small nanophytoplankton.** Contribution of **a, c, e** picophytoplankton (cells < 3 μm in diameter) and **b, d, f** small nanophytoplankton (3–10 μm) to total phytoplankton carbon in unamended versus nutrient-enriched treatments at $t = 96$ h. Picophytoplankton carbon was calculated from flow cytometry measurements of cell abundance, whereas the biomass of large nano- and microphytoplankton (cells > 10 μm) was calculated from abundance and biovolume measurements with the FlowCAM. The carbon of small nanophytoplankton was estimated as the difference between total phytoplankton carbon and the sum of picophytoplankton and large nano- and microphytoplankton carbon. For unamended treatments, total phytoplankton C was estimated from Chl $a$ concentration by applying a C:Chl $a$ ratio (g:g) of 137 (see ref. [51]). For nutrient-enriched treatments, we used a C:Chl $a$ ratio of **a, b** 120, **c, d** 100, and **e, f** 80 g:g. See the "Methods" section for details.

reactions of photosynthesis[61]. Increasing chlorophyll content under warmer temperatures allows cells to strike a balance between the temperature-dependent dark reactions and the largely temperature-independent light reactions that supply energy and reducing power to be used in carbon fixation[22,62]. Accordingly, we found that warming alone enhanced the cellular fluorescence of all three picophytoplankton groups in most of our experiments. The warming- and nutrient-driven enhanced resource allocation into photosynthetic catalysts reinforce each other, hence the largest biomass response in our experiments was observed in the warmed and nutrient-enriched treatment at the warmest and least oligotrophic location (12.7 °C).

A novel finding from our study is that the biomass response of tropical and subtropical phytoplankton assemblages to increased nutrient availability is largely independent of temperature over a 6 °C range. In particular, the response to added nutrients in samples subjected to a 3 °C cooling is relevant to understand the biological impacts of tropical storms. The passage of a cyclone or a typhoon can cause a cooling of sea surface temperature of 1–6 °C over a period of hours to days[63,64]. While the increased surface Chl $a$ typically observed during and immediately after these events may result not only from enhanced in situ growth due to nutrient injection[47,65] but also phytoplankton entrainment

into the surface layer[66], our observations of a similar biomass response to added nutrients in cooled samples and in the control suggest that a sudden decrease in temperature does not hinder the ability of tropical phytoplankton to respond to nutrient injections.

The relatively small effect of temperature on the net growth response of the tropical phytoplankton assemblages investigated here is contrary to both theoretical predictions[67,68] and laboratory-based assessments of the temperature sensitivity of phytoplankton in general[15,18], and picophytoplankton in particular[35,36]. However, it agrees with the observation that nutrient-limited growth conditions greatly reduce the effect of temperature on phytoplankton metabolic rates[20,22,23]. In most laboratory experiments designed to characterize phytoplankton thermal response curves, nutrient concentrations are at least three orders of magnitude higher than those typically found in surface waters of the open ocean. But low nutrient availability, prevailing in most of the open ocean, leads to low intracellular substrate concentrations for biosynthetic reactions, and under these conditions, the temperature-dependence of enzymatic half-saturation constants becomes more significant than that of maximum reaction rates[69]. If both maximum reaction rates and half-saturation constants increase with temperature at the same pace,

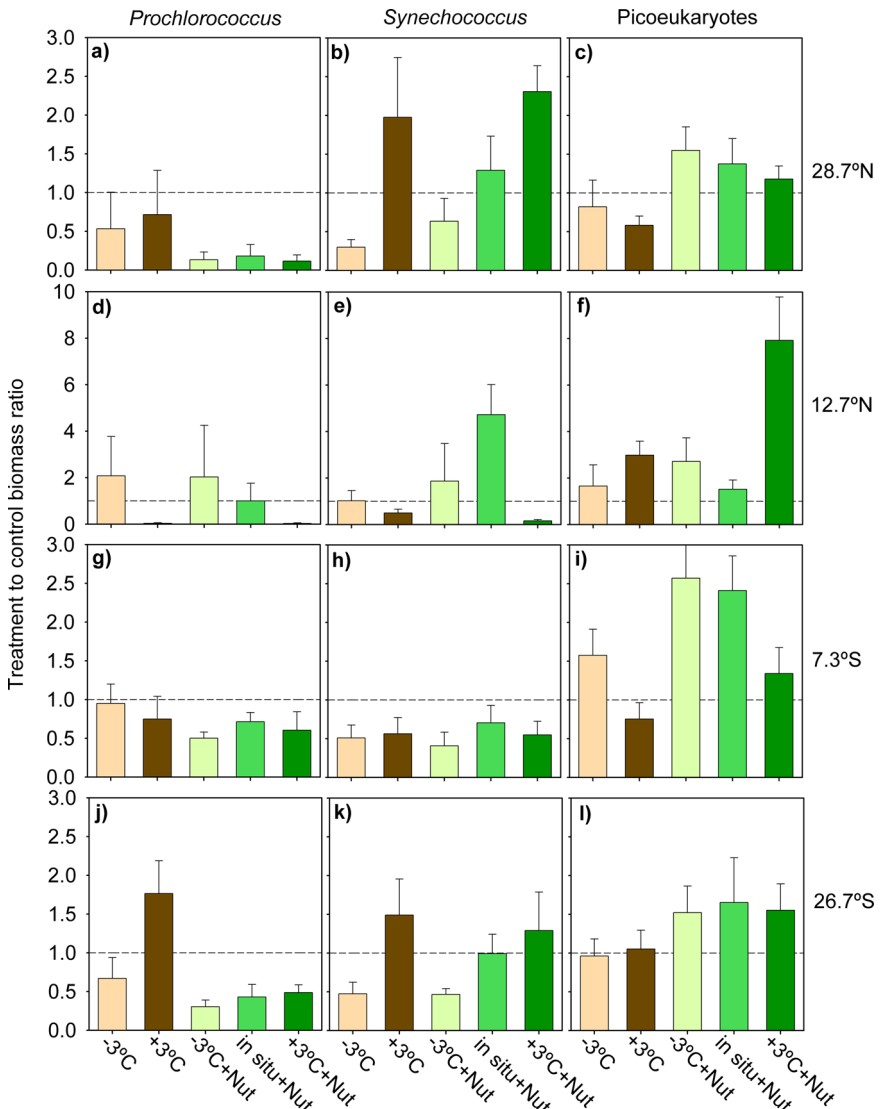

**Fig. 6 Biomass response of *Prochlorococcus*, *Synechococcus* and picoeukaryotes.** Biomass response ratio calculated as the biomass in a given treatment divided by the biomass in the control (in situ temperature and no nutrient addition) at $t = 96$ h for **a**, **d**, **g**, **j** *Prochlorococcus*, **b**, **e**, **h**, **k** *Synechococcus* and **c**, **f**, **i**, **l** picoeukaryotes in the experiments conducted at **a**–**c** 28.7°N, **d**–**f** 12.7°N, **g**–**i** 7.3°S and **j**–**l** 26.7°S. Brown and green represent treatments without and with nutrient additions, respectively. Panels are arranged in columns and rows that correspond to the different picophytoplankton groups and experiments, respectively. Error bars show the standard deviation calculated by the propagation of uncertainty. The dashed line indicates a treatment-to-control ratio of 1. Note differences in the *y*-axis scale.

the two processes counterbalance each other resulting in little temperature sensitivity of metabolic rates and growth[20,69,70]. It is also possible that grazing activity may have been stimulated in the warmed treatments, thus contributing to the lack of a sizable net increase in phytoplankton biomass[71]. However, nutrient addition did cause increases in picophytoplankton abundance and total Chl *a* in all experiments (after a lag time of 24–48 h), which means that grazing pressure was not sufficient to check phytoplankton enhanced growth rates.

The short duration of our experiments may have prevented full acclimation of individuals and populations to changed temperature conditions and thus contributed to the relatively small response of phytoplankton biomass to warming and cooling. Conversely, sudden temperature changes can exacerbate negative effects on growth, particularly in the case of warming, precisely because thermal acclimation requires longer exposure times[72]. In any case, unequivocal phytoplankton growth responses to changes in temperature have been observed over time scales ≤ 4 days

both in laboratory cultures[73] and in natural assemblages of coastal regions[74,75] as well as the open ocean[24]. We, therefore, submit that the comparatively small temperature effects we observed are not due to the short incubation time, but reflect the intrinsic low-temperature sensitivity of phytoplankton growth in oligotrophic regions. This pattern is in agreement with recent experimental and observational evidence indicating the limited direct effect of temperature on phytoplankton growth and productivity in oligotrophic waters of the Red Sea[76] and the tropical Indian Ocean[77]. However, oligotrophic conditions may induce delayed responses to environmental variability and therefore additional experiments using longer incubation times are required to fully characterize the effect of changing temperature on tropical phytoplankton.

Our experimental data, obtained in contrasting locations across the tropical and subtropical Atlantic, do not support the hypothesis that phytoplankton living in the warmest regions are most vulnerable to temperature increases[43]. Indeed, the largest

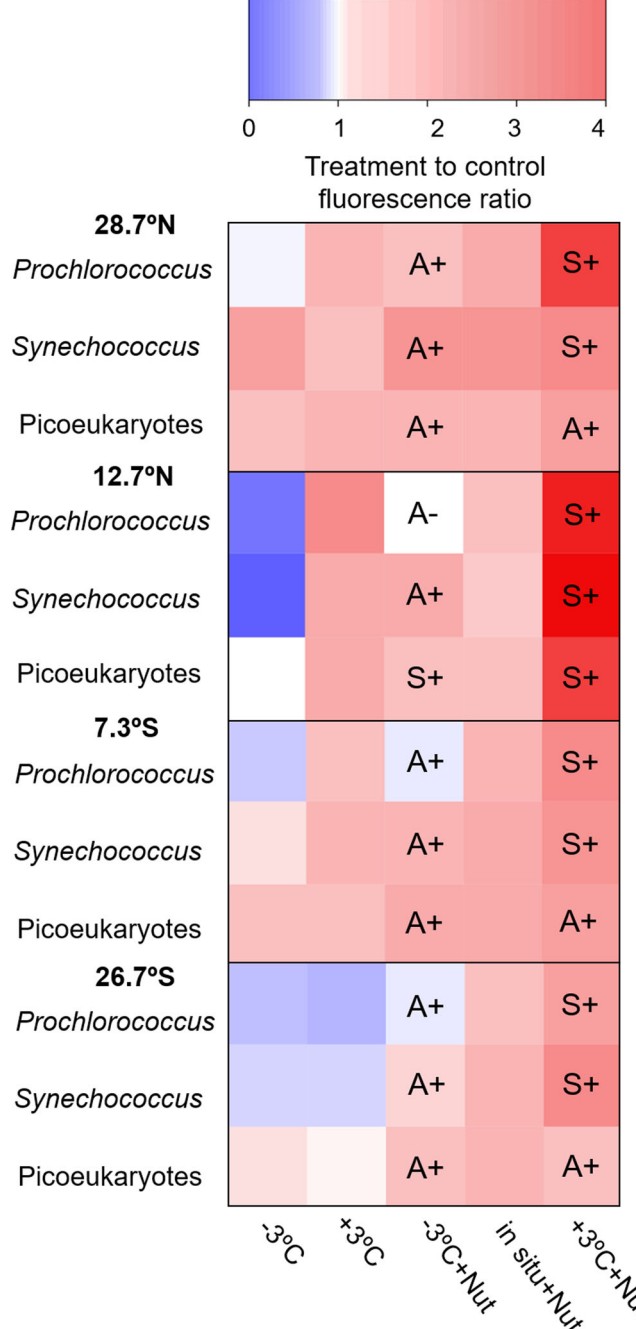

**Fig. 7 Effects of temperature–nutrient treatments on picophytoplankton cellular fluorescence.** Response of *Prochlorococcus*, *Synechococcus* and picoeukaryote cellular red fluorescence to each treatment on each experiment. The response was calculated relative to the control treatment (in situ temperature without nutrient addition) at $t = 96$ h. ANOVA showed significant differences in all cases and Dunnett post-hoc test was performed to assess differences between each treatment and the control (see Supplementary Table 4). Superimposed letters indicate the nature (A, antagonistic; S, synergistic) and direction of the interaction between warming/cooling and nutrient addition. See Supplementary Table 6 for the results of 2-way ANOVA conducted to assess temperature–nutrient interactive effects.

responses to added nutrients, in terms of increased Chl *a*, picophytoplankton biomass, and cell-specific chlorophyll fluorescence, were observed at the warmest location (12.7°N, SST = 28.4 °C) under 3 °C warming, which resulted in an experimental temperature above 32 °C. At this site, there was no negative effect of warming alone on final phytoplankton biomass. These results suggest that thermal response curves obtained in the laboratory, perhaps due to limited genetic variability in cultured strains, may underestimate the potential of phytoplankton to acclimate to sudden environmental changes. Ecosystem nutrient status may be a more relevant factor than temperature to predict the ability of phytoplankton to respond to increased nutrient availability, given that the largest responses to added nutrients were observed in the least oligotrophic locations, as reported before both for the tropical Atlantic[41] and the tropical Pacific[58]. This pattern may result from higher cellular nutrient quotas in regions with less severe oligotrophy[78], and also from the fact that those regions harbour more fast-growing species with the potential to respond to increased nutrient availability[58].

The phytoplankton response to nutrient enrichment was much larger in terms of total Chl *a* (a proxy for bulk phytoplankton biomass) than in terms of picophytoplankton carbon. Moreover, our estimates of the biomass contribution by different size classes indicated that, although picophytoplankton carbon did increase in response to added nutrients, the response of the small nanophytoplankton was stronger, because their contribution to total carbon consistently increased in nutrient-enriched treatments. Thus it seems that most of the net biomass increase after nutrient addition was due to small nanophytoplankton cells, which have been shown to have faster maximum growth rates than their smaller counterparts[37,79]. An enhanced biomass contribution by small nanophytoplankton as a result of experimental nutrient enrichment also agrees with the large-scale patterns identified in the Atlantic Ocean, whereby nanophytoplankton biomass contribution increases from oligotrophic to mesotrophic regions[80,81]. Within the picophytoplankton, picoeukaryotes (2–3 μm in cell diameter) are more responsive to nutrient enrichment than picocyanobacteria (0.5–1.5 μm)[39,42], a pattern related to the positive relationship between cell size and maximum growth rates observed for cells < 5 μm[37,79,82]. Conversely, *Prochlorococcus* has a streamlined genome and reduced nutrient requirements[83] that make it particularly well-adapted to ultraoligotrophic conditions but tends to be outcompeted by larger, faster-growing species when nutrient limitation is relieved, in addition to suffering stronger losses to grazing[28,84]. In agreement, the abundance of *Prochlorococcus* tended to decrease after nutrient addition in our bioassays. In contrast to these responses in nutrient-enriched treatments, we did not find consistent effects of temperature, either on size-fractionated biomass or in the abundance of the different picophytoplankton groups.

In conclusion, our results show that moderate changes in nutrient concentration have a much larger potential than varying temperatures over a 6 °C range to alter the growth and community structure of phytoplankton in the tropical and subtropical Atlantic Ocean. While oligotrophic conditions are associated with reduced temperature sensitivity of phytoplankton growth, sea surface warming may stimulate the ability of phytoplankton assemblages to respond to transient events of enhanced nutrient supply. Thus the interaction between temperature and nutrient supply can be an important factor to predict the effect of climate change on the productivity of the oligotrophic ocean.

## Methods
Samples and data were collected during the 29th Atlantic Meridional Transect (AMT) research cruise (DY110) aboard the RRS Discovery, which took place between Southampton (UK) and Punta Arenas (Chile) from October 13 to November 25, 2019 (Fig. 1a). Along the transect, four locations were selected to investigate experimentally the responses of natural phytoplankton assemblages to simultaneous changes in temperature and nutrient availability.

**Water sampling and hydrographic data collection**. Sampling was conducted before dawn from surface waters (2–5 m) using a rosette with 24 Niskin bottles attached. At each sampling location (28.7°N on October 26, 12.7°N on November 1, 7.3°S on November 7, and 26.7°S on November 12), seawater was transferred directly from the Niskin bottles into a 20-L dark carboy. Vertical profiles of temperature, salinity and chlorophyll fluorescence were obtained with a CTD probe (SeaBird, SBE, 911plus/917). We used the equation of Morel et al.[85] to estimate the 1% PAR depth (1% PAR$_z$) from the depth of the deep chlorophyll maximum. Density ($\sigma_t$) was calculated from temperature and salinity by using the standard UNESCO equation[86]. Mean surface chlorophyll $a$ concentration data from NASA Aqua-MODIS for the period October–November 2019 (Fig. 1a) were downloaded from https://oceancolor.gsfc.nasa.gov/l3/order/. For the analysis of dissolved inorganic nutrients, 60-mL samples were collected from the Niskin bottles into acid-washed HDPE (Nalgene) bottles. Clean sampling and sample handling for the determination of dissolved nutrient concentration was carried out according to GO-SHIP protocols[87]. Nutrient samples were frozen at −20 °C immediately after collection until analysis on land. Defrosting was conducted according to the procedures outlined in Becker et al.[87] and dissolved nutrient concentrations were determined using a SEAL analytical AAIII segmented-flow, colorimetric nutrient auto-analyser. The detection limit of these analyses was 0.01 µmol L$^{-1}$ for nitrite, 0.02 µmol L$^{-1}$ for nitrate + nitrite and 0.02 µmol L$^{-1}$ for phosphate.

The depth of the nitracline was set as the first depth at which the concentration of nitrate was higher than 0.5 µmol L$^{-1}$. The depth of the deep chlorophyll maximum (DCM$_z$) was taken as the depth of the deep fluorescence peak. We calculated the resource supply index (RSI), following Marañón et al.[88], to quantify differences in light and nutrient availability among locations:

$$\text{RSI} = \frac{[\text{NO}_3^- + \text{NO}_2^-]_{1\%\text{PAR}_z}}{\Delta\sigma_t} \times \frac{1\%\text{PAR}_z}{\text{UML}_z} \tag{1}$$

where $[\text{NO}_3^- + \text{NO}_2^-]_{1\%\text{PAR}_z}$ is the nitrate plus nitrite concentration at the 1% PAR depth, $\Delta\sigma_t$ is the density difference between the surface and the base of the euphotic zone, 1%PAR$_z$ is the depth at which irradiance equals 1% of incident PAR, and UML$_z$ is the depth of the upper mixed layer, estimated as the depth at which $\sigma_t$ is 0.125 units higher than at the surface.

**Temperature–nutrient treatments and experimental setup**. Three temperature treatments, a warmer (+3 °C relative to in situ), an in situ (no change in temperature) and a colder temperature (−3 °C relative to in situ) were crossed with two nutrient treatments, one with the addition of NH$_4$NO$_3$ and NaH$_2$PO$_4$ to give a concentration increase of 1 µmol L$^{-1}$ for nitrate and ammonium and 0.2 µmol L$^{-1}$ for phosphate, and the other with in situ nutrient concentration (unamended). The magnitude of the N and P amendments was chosen to coincide with earlier nutrient bioassays in tropical, oligotrophic waters[26,28,50,58]. The experimental setup resulted in six treatments, each of which was performed in triplicate for a total of 18 experimental units (1-L polycarbonate bottles). Nutrient stock solutions were prepared with pre-weighted (on land) AnalaR-grade reagents by dilution in Milli-Q water prior to the first experiment. Samples to verify the initial nutrient enrichment were collected at the start of each experiment (see Supplementary Table 7).

To control temperature, we used a specially designed incubator (called 'Planktotherm', Supplementary Fig. 4) that was located on the ship's aft deck. This incubator is equipped with three cylindrical polycarbonate tanks of 100 L in volume with internal water recirculation and distribution at 5 heights to avoid blind spots of temperature or stratification. All tanks had a polycarbonate cap that limited the loss of water in rough sea conditions or through evaporation. The temperature was monitored continuously by the internal temperature-controlling system of the Planktotherm as well as by submerged TinyTag data loggers (TinyTag Aquatic 2) that recorded the temperature every 2 min (Supplementary Fig. 5).

The incubation bottles were acid-washed and rinsed with ultrapure water. Prior to filling them with the incubation sample, they were rinsed twice with natural seawater. Once all 18 bottles were filled, half of them were amended with nutrients as explained above. We then distributed the bottles among the tanks, so that 6 bottles (3 nutrient-enriched and 3 unamended) were placed at each experimental temperature. To prevent photodamage, all bottles were covered by a neutral density mesh that screened out 30% of incident irradiance.

Samples were taken for the determination of initial chlorophyll $a$ concentration (Chl $a$), active chlorophyll $a$ fluorescence (ChlF) parameters, picophytoplankton abundance and large nano- and microphytoplankton abundance. All experiments lasted for 96 h, and every 24 h, just before sunrise, the 18 bottles were sampled for picophytoplankton abundance and ChlF parameters. In addition, at the end of the experiment, Chl $a$ concentration and the abundance of large nano- and microphytoplankton were also determined for each of the 18 bottles.

**Chlorophyll $a$ concentration**. On the first day of incubation, triplicate 200–250 mL samples were taken immediately from the original carboy to measure initial chlorophyll $a$ concentration (Chl $a$). On the last day (at 96 h of incubation), 200–250 mL samples were taken from each of all the 18 bottles. Chl $a$ concentration was measured fluorometrically after filtration of the sample through 0.2-µm polycarbonate filters. The filters were stored in Eppendorf tubes and frozen at −20 °C until analysis. Chlorophyll $a$ was then extracted in 6 mL of 90% HPLC-grade acetone and stored at 5 °C overnight. Chl $a$ concentration was determined with a Trilogy fluorometer (Trilogy Laboratory Fluorometer, Turner Designs) that had been calibrated with pure Chl $a$.

**Active chlorophyll $a$ fluorescence parameters**. Single-turnover chlorophyll $a$ fluorescence (ST-ChlF) measurements were performed daily using a fast repetition rate fluorometer (FRRF, FastOcean, Chelsea Technologies Ltd.). Subsamples were taken from all bottles before sunrise and kept at low light in a temperature-controlled water bath at in situ temperature until processing. At 24, 48 and 72 h, samples from the three replicates of each treatment were pooled together. The initial sample at 0 h was run in triplicate and at 96 h all 18 samples were measured individually. Blanks were run for each sample after gentle filtration through 0.2-µm Acrodisc syringe filters. ST-ChlF induction curves were measured using the standard protocol of 100 excitation flashlets at a 2-µs pitch and 450-nm excitation wavelength. The intensity of the excitation LED and the gain of the photomultiplier was adjusted automatically by the instrument. To increase the signal-to-noise ratio, each induction curve was fitted to 20–50 acquisitions. Samples were maintained in their dark-regulated state and the minimum ($F_o$) and maximum ($F_m$) ChlF were measured to determine $F_v/F_m$:

$$F_v/F_m = (F_m - F_o)/F_m \tag{2}$$

$F_v/F_m$ is an estimate of the maximum quantum yield of photochemistry in photosystem II[89]. $F_o$ was strongly correlated to extracted Chl $a$, so that the following linear regression model ($R^2 = 0.867$, $F(1, 89) = 580.22$, $p < 0.01$, $n = 90$) was obtained:

$$\text{Chl}a = 0.383F_o + 0.022 \tag{3}$$

Equation (3) was used to estimate Chl $a$ concentration from $F_o$ for all samples throughout the experiment and thus characterize the temporal evolution. However, when only initial and final Chl $a$ concentrations are considered (at $t = 0$ and $t = 96$ h), the Chl $a$ concentration reported corresponds to actual measurements of extracted Chl $a$ (i.e. not estimated from $F_o$).

**Picophytoplankton abundance and biomass**. Every 24 h we took 6-mL samples from the 18 incubation bottles to determine the abundance of the cyanobacteria *Prochlorococcus* and *Synechococcus* and the picoeukaryotic algae (cells < 3 µm in diameter). Samples were taken in dark Falcon tubes$^{\text{TM}}$, transported into the lab immediately and kept in the dark at 4 °C until analysis by analytical flow cytometry following Tarran et al.[81]. Briefly, we used a Becton Dickinson FACSort$^{\text{TM}}$ (BD Biosciences) equipped with an air-cooled laser providing blue light at 488 nm. Daily flow rate calibrations were made prior to the analysis of the samples using Beckman Coulter$^{\text{TM}}$ Flowset$^{\text{TM}}$ fluorospheres at a known concentration. In addition to counting cells, the flow cytometer also measured chlorophyll red fluorescence (>650 nm), phycoerythrin fluorescence (585 ± 21 nm), and side scatter, which is the light scattered at 90° to the direction of the laser beam. The red fluorescence signal was used as a proxy for chlorophyll $a$ cellular content. Data of light scatter and fluorescence were processed using CellQuest software (Becton Dickinson, Oxford) with log amplification on a four-decade scale with 1024-channel resolution. Scatter plots of side scatter vs. orange fluorescence were used to discriminate and enumerate *Synechococcus*, and plots of side scatter vs. red fluorescence (without *Synechococcus*) were used to count *Prochlorococcus* and picoeukaryotic algae. To estimate carbon biomass from cell abundance we used the conversion factors obtained by Zubkov et al.[30] in the central Atlantic Ocean: 32 fg C cell$^{-1}$ for *Prochlorococcus*, 103 fg C cell$^{-1}$ for *Synechococcus* and 1496 fg C cell$^{-1}$ for picoeukaryotes.

**Nano- and micro-phytoplankton abundance and biomass**. The abundance and cell volume of large nano- and micro-phytoplankton (cells > 10 µm in equivalent spherical diameter, ESD) were determined by automated digital imaging. Two samples (30 mL in volume) were collected at the beginning of the experiment to determine initial abundance and samples from all incubation bottles were taken at $t = 96$ h. All samples were fixed with 3 mL of buffered-basic pH formaldehyde 4% (1% final concentration) and stored at room temperature in the dark until analysis on land. Samples were examined with a FlowCAM® 8400 (Yokogawa Fluid Imaging Technologies, Inc.) instrument and images were processed with the paired particle analysis software VisualSpreadsheet©. Prior to imaging, the 30-mL samples were concentrated into 5 mL by sedimentation for 24 h in Utermöhl chambers. Samples were run using a flow cell of 100 µm field of view at ×10 magnification with a flow rate of 100 µL min$^{-1}$. Once the sample was photographed, images were identified by the software and categorized by taxonomic groups (dinoflagellates, diatoms, silicoflagellates and protozoa). Particle dimensions and biovolumes were then estimated for each identified organism. To compute carbon biomass per cell ($C$, pg C cell$^{-1}$) from biovolume ($V$, µm$^3$) we used the empirical function $C = 0.22 V^{0.88}$ (see ref. [37]). We estimated the carbon biomass of small nanophytoplankton (cells with ESD between 3 and 10 µm) as the difference between total carbon biomass and the carbon biomass contributed by picophytoplankton and the large nano- plus microphytoplankton size classes, derived from flow cytometry and FlowCAM data, respectively. For initial samples and treatments without added nutrients, total phytoplankton carbon biomass was calculated from Chl $a$ by applying the mean C:Chl $a$ ratio (137 g:g) measured in surface oligotrophic waters

of the tropical and subtropical Atlantic Ocean[51]. Using a different value for the initial C:Chl $a$ ratio yields a different estimate of total phytoplankton carbon, but it does not affect the assessed impact of experimental treatments upon the size-partitioning of phytoplankton biomass. Given that Chl $a$ cellular content increases when nutrient availability is enhanced, for the nutrient-amended treatments we applied C:Chl $a$ ratios of 120, 100 and 80 g:g, which represent different degrees of nutrient-induced pigment increase. We used the empirical model of Geider[61], based on laboratory measurements with cultures of numerous species (including diatoms, chlorophytes and cyanobacteria) under different temperature and irradiance conditions, to estimate the C:Chl $a$ ratios that can be expected under the high-light conditions of tropical and subtropical regions. The mean daytime incident PAR during our experiments was approximately 260 W m$^{-2}$ (see Supplementary Data 3), which corresponds to a photon flux density of 1200 µmol m$^{-2}$ s$^{-1}$. For a PAR of 1200 µmol m$^{-2}$ s$^{-1}$ and a temperature range of 20–28 °C, the model of Geider[61] predicts a C:Chl $a$ range of 77–199 (g:g) under nutrient-replete conditions. Given the oligotrophic conditions prevailing during our study, it is unlikely that the investigated phytoplankton assemblages ever reached C:Chl $a$ ratios below 80. Finally, once the carbon biomass of each size class was computed, we assessed the effect of nutrient addition on size structure by plotting the contribution of pico- and small nanophytoplankton to total biomass in unamended bottles at each temperature against the contribution observed under nutrient-enrichment at the same temperature.

**Statistics and reproducibility.** We used two-tailed Student's $t$-tests to assess differences in total Chl $a$ concentration and picophytoplankton biomass between treatments with added nutrients (+Nut) and treatments without nutrient amendment using data from the four experiments. We applied one-way analysis of variance, ANOVA, to test for differences among the different temperature and nutrient treatments in group-specific picophytoplankton biomass and cellular red fluorescence, using the original, untransformed data from each experiment. In the ANOVA, temperature and nutrient availability were considered as independent variables with 3 and 2 levels, respectively, while carbon biomass and chlorophyll fluorescence were considered as dependent variables. Post-hoc Dunnett tests were conducted to test the differences in biomass and fluorescence between each of the five treatments (−3, +3 °C, −3 °C +Nut, in situ +Nut, +3 °C +Nut) and the control (in situ temperature, no nutrient addition). We also conducted 2-way ANOVA tests to assess the effect of temperature, nutrients and the temperature × nutrient interaction on picophytoplankton biomass and cellular fluorescence on each experiment. The method of Piggott et al.[90] was used to classify the type (synergism, antagonism) and direction (positive, negative) of the temperature × nutrient interaction. All data were checked with a homoscedasticity Levene's test as well as with a normality Shapiro–Wilks's test. All statistical analyses were carried out with SPSS v.22 (IBM, Armonk, NY, USA) and R studio v 4.0.3 (RStudio, Boston, MA, USA). Matlab (The MathWorks, Inc., Natick, MA, USA) and Sigma Plot v.10.0 (SysTat Software, Berkshire, UK) were also used to plot graphs.

**Reporting summary.** Further information on research design is available in the Nature Research Reporting Summary linked to this article.

## Data availability

Data are archived at the British Oceanographic Data Centre (BODC): https://www.bodc.ac.uk/data/published_data_library/catalogue/10.5285/d78073fe-dc77-7d2f-e053-6c86abc06500/

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

## Acknowledgements

This research was supported by the Spanish Ministry of Science and Innovation through research grant PGC2018-094553-B-I00 and by the European Union through H2020 TRIATLAS project 'Tropical and South Atlantic climate-based marine ecosystem predictions for sustainable management' (Grant agreement No. 817578). C.F.-G. acknowledges the receipt of a predoctoral research fellowship from Xunta de Galicia (ED481-2017/342). We thank Rebecca May and Paul Strubinger for collecting nutrient samples at

sea, Giorgio Dall'Olmo for irradiance data, Minerva Espino for plankton analysis with the FlowCAM, and Esperanza Broullón for advice with Matlab. We are also grateful to principal scientist Giorgio Dall'Olmo and the captain and crew of RRS Discovery for their support during the work at sea. The Atlantic Meridional Transect is funded by the UK Natural Environment Research Council through its National Capability Long-term Single Centre Science Programme, Climate Linked Atlantic Sector Science (grant number NE/R015953/1). This study contributes to the international IMBeR project and is contribution number 379 of the AMT programme.

## Author contributions

E.M. designed the research. C.F.-G. carried out the experiments at sea and obtained data. G.A.T., N.S., E.M.S.W. and J.A. obtained data. C.F.-G. and E.M. analysed the data and wrote the paper. All authors commented on the article.

## Competing interests

The authors declare no competing interests.
