## [Peer Review File · Communications Biology]

Reviewers' comments:

Reviewer #1 (Remarks to the Author):

Through two-factor microcosm experiments, this paper investigates the interactive effects of temperature and nutrient supply on phytoplankton growth in the tropical and subtropical Atlantic. The significant findings of the studies are:

1. Nutrient availability plays a much larger role than temperature as a driver of phytoplankton growth dynamics.
2. Different picophytoplankton groups exhibit contrasting responses to nutrient enrichment.
3. Picophytoplankton cellular chlorophyll content is stimulated by both increased temperature and nutrient addition.

In particular, conclusion (i) has been shown in laboratory settings but has not been previously reported on a large spatial scale. These findings are exciting and worthy of consideration for publication.

Overall, I am satisfied with the quality of work presented, and the data in this manuscript are convincing and presented by well-designed experiments. The manuscript is well written, and the results are of sufficient quality. I have two amends to suggest regarding the statistical analyses of data (see general comments). Addressing these concerns would strengthen the conclusions of the manuscript.

General comments:

(1) The treatment of 'ratio' as effect size: In Figures 4, 6, and 7, the authors use ratios (biomass and fluorescence) as an effect size between control and treatment and then conduct ANOVA to show significant differences. However, by their nature, ratios do not respond symmetrically to changes in the numerator and denominator and do not follow normal distributions (for example, Isles, P. D. (2020). The misuse of ratios in ecological stoichiometry. *Ecology*, 101(11), e03153.). For non-normally distributed outcomes such as ratios, parametric tests like ANOVA are inappropriate.

To alleviate these issues with ratios, statisticians recommend using log-transformed ratio = $\log(\text{treatment}/\text{control})$. This log-transformed ratio is appropriate for parametric tests such as ANOVA because the log ratio of zero corresponds to the true absence of the outcome and is normally distributed. I, therefore, urge authors to conduct ANOVA using log-transformed ratios. As for figures, I think either simple or log-transformed ratios are acceptable, and the former may be more intuitive for interpretation.

(2) Need for more rigorous analysis of interactive effects between nutrients and temperature: On a related note, there is a statistical technique to rigorously test whether the interactive effects of temperature and nutrients are either additive, synergistic, or antagonistic. The way the authors currently treat and describe interactive effects is qualitative and does not give concrete statistical evidence backing up their claims, for example, that "temperature and nutrient addition synergistically enhanced fluorescence." (Line 520-521).

To determine the nature of interactive effects (additive, synergistic, or antagonistic), I suggest authors refer to papers such as the following:

* Piggott, J. J., Townsend, C. R., & Matthaei, C. D. (2015). Reconceptualizing synergism and antagonism among multiple stressors. *Ecology and Evolution*, 5(7), 1538-1547.

* Crain, C. M., Kroeker, K., & Halpern, B. S. (2008). Interactive and cumulative effects of multiple human stressors in marine systems. *Ecology Letters*, 11(12), 1304-1315.

Specific comments:

1. Line 14 on Data Availability Statement: Could authors be more specific about where the data are available (DOI, metadata, etc.)?
2. Lines 58-59: "there is growing evidence that the temperature sensitivity of phytoplankton growth and metabolism is reduced when nutrient availability is low." - Authors need to provide

relevant references.

3. Line 105: Should be "the ones that benefit least" not "benefit less."

4. Line 156: Check that subscripts for NO₃⁺ are correctly formatted.

5. Line 399: The latitudes in legends are in integers (13N and 7S), but it is written in 1 decimal place in the text. Authors should make the formatting of latitude consistent in the figures and the main text.

6. Figure 4: Useful to put a letter symbol to indicate the significant difference between treatments. Also, the y-axis should be something more informative such as "ratio (Biomass treatment/Biomass control)."

7. Figure 7: Again y-axis should be more informative than just the "ratio" to distinguish the fluorescence-based ratio from the biomass-based ratio in Figures 4 and 6.

8. Line 526: also could cite [UstICK, L. J., Larkin, A. A., Garcia, C. A., Garcia, N. S., Brock, M. L., Lee, J. A., ... & Martiny, A. C. (2021). Metagenomic analysis reveals global-scale patterns of ocean nutrient limitation. *Science*, 372(6539), 287-291.]

9. Line 649-653: Could authors elaborate on what they mean by "positive size-scaling maximum growth rates for cells < 5 μm". How is the size difference between picocyanobacteria and picoeukaryotes related to growth rates?

reviewed by: Tatsuro Tanioka

Reviewer #2 (Remarks to the Author):

In this study, the authors explore the combined effect of changes in temperature and nutrient availability on the dynamics of picophytoplankton groups through an experiment using in situ communities in the Atlantic Ocean. The authors selected different zones along the AMT transect with different conditions to investigate the response of these groups to temperature and nutrient variability.

Overall, this work is very interesting and timely, as it addresses a key issue that has many implications in the current climate change situation. The study site selected to conduct the experiment is very convenient to address the question and relevant, as includes one of the most important oligotrophic regions of the ocean, but also a latitudinal transect that includes different in situ conditions.

The experiment is well designed and reported, and the lab analyses are well performed and described. I cannot evaluate nutrient analysis or active chlorophyll a method as I don't have experience with them.

Major concerns

1. The statistical analysis, in my opinion, needs to be revised and/or better reported. It is difficult to evaluate what they have done following the manuscript as it stands. For this reason, some of their statements are not well supported. To be clear, I don't think there is anything wrong with them, but they need to be better reported and, in my opinion the analyses are not sufficient to support some of the statements (see detailed comments below).

In addition, the authors should provide more information about the test applied, what type of ANOVA or t-test was used, the variable included and how it was included. What kind of information does the test provide? The threshold used to interpret the tests. I think that it is also crucial to report all the information that the test provides not just the p-value.

Another way to analyze the data could be to write a model accounting for the hierarchical structure of the data to tease apart the effect of temperature and nutrients on the different response variables accounting for the variability of the different sampling sites. I am not sure how feasible this would be with your dataset, but this could help to understand the effect of temperature on e.g., biomass, the effect of nutrient addition on biomass, and the effect of the interaction between the two factors on biomass accounting for the variability of each station.

2. A concern I have is the time scale of the study, especially with regard to the assessment of the effect of temperature. I think the authors could explain or acknowledge that this study was conducted on a very short time scale and the implications this could have on the results obtained. My experience addressing the temperature dependence of metabolism and growth using different cyanobacteria and phytoplankton species suggests that the species need an acclimation at each treatment before measuring these variables when possible and, even if not possible, the time scale used is several weeks. I am aware that this is not feasible on board. But do you think the lower response of the species to temperature, especially without adding nutrients, could be related to the short duration of the experiment?

Detailed comments

L51. Could you use a more actual reference here?

L141. Please add a reference / equation for the UNESCO equation.

L248. The flow cytometer measures red fluorescence which is a proxy of the chlorophyll content of the cell.

L254 is there any reason to remove *Synechococcus* here? Do they overlap?

L322 Could you please report here the test you used, and p-value obtained?

L354-357 I understand that figure 2 is useful to see general pattern, but I disagree with the authors that the dynamics shown in Fig S3 and S4 are similar in the 4 experiments. The response is quite different between treatments and not all the experiments respond to the treatments at the same time (e.g., Fig S4a). Could you please clarify this statement?

L358 In line with my previous comment. Although figure 2 is useful as an overview, I think it is also good to keep in mind figure S4 and report the exceptions.

L405-407 Are you referring to temperature treatments without nutrient addition? What role do you think incubation time plays in this response?

Figure 4 and 6 Could you change the color coding here? I think it is confusing to use colors associated with temperature here.

L447 How did the authors verify this?

L516-517 I am not convinced by this statement. First, because the authors do not show any estimation of growth data in the analysis. The authors mainly test the biomass at the end of the experiment. Second, I am not convinced that the tests presented here can quantify this. They tested whether one treatment was different from another, but not the effect of nutrients and temperature alone. Third, I believe that incubation time here might play a role in the temperature response of these organisms, especially under nutrient-limited conditions. Therefore, in my opinion, I think the authors should either soften the statement in this regard or acknowledge the limitations of the study.

L540-545 I agree with this paragraph, but I suggest including a few sentences about how the response time might be longer in case of temperature change.

L548 typo temperature

L581-582 I am not convinced that the authors may claim this based on the results or at least clarify that this response is in short time scale (less than a week). The authors observe an important increase in biomass when combining temperature and nutrients but also differences between temperature treatments.

Response to the referees' comments

Reviewer #1 (Remarks to the Author):

Through two-factor microcosm experiments, this paper investigates the interactive effects of temperature and nutrient supply on phytoplankton growth in the tropical and subtropical Atlantic. The significant findings of the studies are:

1. Nutrient availability plays a much larger role than temperature as a driver of phytoplankton growth dynamics.
2. Different picophytoplankton groups exhibit contrasting responses to nutrient enrichment.
3. Picophytoplankton cellular chlorophyll content is stimulated by both increased temperature and nutrient addition.

In particular, conclusion (i) has been shown in laboratory settings but has not been previously reported on a large spatial scale. These findings are exciting and worthy of consideration for publication.

Overall, I am satisfied with the quality of work presented, and the data in this manuscript are convincing and presented by well-designed experiments. The manuscript is well written, and the results are of sufficient quality. I have two amends to suggest regarding the statistical analyses of data (see general comments). Addressing these concerns would strengthen the conclusions of the manuscript.

We are very grateful to Dr. Tatsuro Tanioka for his constructive and detailed review. The suggestions regarding the statistical treatment of interactive effects have been especially useful. We provide below specific responses to each point raised.

General comments:

(1) The treatment of 'ratio' as effect size: In Figures 4, 6, and 7, the authors use ratios (biomass and fluorescence) as an effect size between control and treatment and then conduct ANOVA to show significant differences. However, by their nature, ratios do not respond symmetrically to changes in the numerator and denominator and do not follow normal distributions (for example, Isles, P. D. (2020). The misuse of ratios in ecological stoichiometry. *Ecology*, 101(11), e03153.). For non-normally distributed outcomes such as ratios, parametric tests like ANOVA are inappropriate.

To alleviate these issues with ratios, statisticians recommend using log-transformed ratio = $\log(\text{treatment}/\text{control})$. This log-transformed ratio is appropriate for parametric tests such as ANOVA because the log ratio of zero corresponds to the true absence of the outcome and is normally distributed. I, therefore, urge authors to conduct ANOVA using log-transformed ratios. As for figures, I think either simple or log-transformed ratios are acceptable, and the former may be more intuitive for interpretation.

Indeed we are aware that ratios are not appropriate metrics to be used in ANOVA. We clarify that although ratios were shown in the figures, they were not used in the ANOVA tests, which were conducted using the original, untransformed biomass data. In order to avoid any misunderstanding, we have re-written the relevant passages in the

‘Statistical analyses’ section, which explain also the additional analyses conducted (lines 671-683):

“We applied one-way analysis of variance, ANOVA, to test for differences among the different temperature and nutrient treatments in group-specific picophytoplankton biomass and cellular red fluorescence, using the original, untransformed data. In the ANOVA, temperature and nutrient availability were considered as independent variables with 3 and 2 levels, respectively, while carbon biomass and chlorophyll fluorescence were considered as dependent variables. Post-hoc Dunnett tests were conducted to test the differences in biomass and fluorescence between each of the five treatments (-3°C , $+3^{\circ}\text{C}$, $-3^{\circ}\text{C} + \text{Nut}$, *in situ* $+ \text{Nut}$, $+3^{\circ}\text{C} + \text{Nut}$) and the control (*in situ* temperature, no nutrient addition). We also conducted 2-way ANOVA tests to assess the effect of temperature, nutrients and the temperature \times nutrient interaction on picophytoplankton biomass and cellular fluorescence. The method of Piggott et al. (2015) was used to classify the type (synergism, antagonism) and direction (positive, negative) of the temperature \times nutrient interaction”.

The legends to Tables S2 and S3 have also been modified to make it clear that the original, untransformed data were used in the ANOVA tests.

(2) Need for more rigorous analysis of interactive effects between nutrients and temperature: On a related note, there is a statistical technique to rigorously test whether the interactive effects of temperature and nutrients are either additive, synergistic, or antagonistic. The way the authors currently treat and describe interactive effects is qualitative and does not give concrete statistical evidence backing up their claims, for example, that “temperature and nutrient addition synergistically enhanced fluorescence.” (Line 520-521).

To determine the nature of interactive effects (additive, synergistic, or antagonistic), I suggest authors refer to papers such as the following:

* Piggott, J. J., Townsend, C. R., & Matthaei, C. D. (2015). Reconceptualizing synergism and antagonism among multiple stressors. *Ecology and Evolution*, 5(7), 1538-1547.

* Crain, C. M., Kroeker, K., & Halpern, B. S. (2008). Interactive and cumulative effects of multiple human stressors in marine systems. *Ecology Letters*, 11(12), 1304-1315.

We greatly appreciate this suggestion and thank the reviewer for providing these references. In the revised manuscript, we now follow the procedure outlined by Piggott et al. (2015) to determine the type and direction of interactive effects between temperature and nutrient availability in our experiments. The type of interaction observed (synergy or antagonism) and its direction (positive or negative) are now indicated on Fig. 7 with superimposed letters and signs.

In addition, following the advice from both reviewers, we have conducted new, 2-way ANOVA tests to assess the interactive effects between temperature and nutrient availability on both picophytoplankton biomass and cellular fluorescence. The results of these new analyses are given in Tables S5 and S6, which are referred to in the Results section. We have nevertheless chosen to maintain in the supplementary information the original 1-way ANOVA and the subsequent post-hoc tests, because they allow to

examine the differences between each individual treatment and the control, in direct correspondence to the results shown in the figures.

Specific comments:

1. Line 14 on Data Availability Statement: Could authors be more specific about where the data are available (DOI, metadata, etc.)?

The DOI is now given.

2. Lines 58-59: “there is growing evidence that the temperature sensitivity of phytoplankton growth and metabolism is reduced when nutrient availability is low.” - Authors need to provide relevant references.

The relevant references are given in the immediately subsequent sentences. That sentence is a general statement about the nutrient-dependence of phytoplankton temperature sensitivity, which is elaborated upon in lines 53-66 of the revised manuscript. The relevant references are: Skau et al. 2017, Marañón et al. 2018, Fernández-González et al. 2020, O’Connor et al. 2009, Marañón et al. 2018, López-Sandoval et al. 2020, Liu et al. 2020, and Hayashida et al. 2020.

3. Line 105: Should be “the ones that benefit least” not “benefit less.”

Changed as suggested.

4. Line 156: Check that subscripts for NO₃⁺ are correctly formatted.

Done.

5. Line 399: The latitudes in legends are in integers (13N and 7S), but it is written in 1 decimal place in the text. Authors should make the formatting of latitude consistent in the figures and the main text.

In the revised manuscript, all latitudes in both text and figures are formatted with 1 decimal place.

6. Figure 4: Useful to put a letter symbol to indicate the significant difference between treatments. Also, the y-axis should be something more informative such as “ratio (Biomass treatment/Biomass control).”

We have changed the y-axis label to the more informative ‘Treatment to control biomass ratio’ in both Figs. 4 and 6. To avoid making binary decisions based on a sharp, cut-off value (see Amrhein et al. 2019, Muff et al. 2022) we prefer not to indicate significant ($p < 0.05$) differences with a symbol in this figure. Readers can use Tables S2 and S3 to examine the strength of the evidence for significant effects of the different treatments.

7. Figure 7: Again y-axis should be more informative than just the “ratio” to distinguish the fluorescence-based ratio from the biomass-based ratio in Figures 4 and 6.

As suggested, we have changed the colour-scale label in Fig. 7 to ‘Treatment to control fluorescence ratio’.

8. Line 526: also could cite [Ustick, L. J., Larkin, A. A., Garcia, C. A., Garcia, N. S., Brock, M. L., Lee, J. A., ... & Martiny, A. C. (2021). Metagenomic analysis reveals global-scale patterns of ocean nutrient limitation. *Science*, 372(6539), 287-291.]

We thank the reviewer for pointing out this study, which is clearly relevant to our manuscript. In the revised version, this study is now cited in the Introduction (line 70) to show that large-scale patterns of nutrient limitation in the Atlantic are being revealed not only through nutrient manipulation experiments but also through metagenomic analysis.

9. Line 649-653: Could authors elaborate on what they mean by “positive size-scaling maximum growth rates for cells < 5 μm ”. How is the size difference between picocyanobacteria and picoeukaryotes related to growth rates?

Observations with both natural populations and laboratory cultures have shown that there is a positive size-scaling of maximum growth rate for cells < 5 μm in equivalent spherical diameter, which means that maximum growth rate increases with increasing cell size. Thus, the maximum growth rate sustained by picoeukaryotic algae (2-3 μm in cell diameter) is higher than that of *Synechococcus* (1-1.5 μm) which, in turn, is higher than that of *Prochlorococcus* (0.5-1 μm). We have re-written the relevant sentence to clarify this point (lines 485-488):

“Within the picophytoplankton, picoeukaryotes (2-3 μm in cell diameter) are more responsive to nutrient enrichment than picocyanobacteria (0.5-1.5 μm) (Mouriño-Carballido et al. 2016; Berthelot et al. 2019), a pattern related to the positive relationship between cell size and maximum growth rates observed for cells < 5 μm in diameter”

References (not cited in the manuscript)

Amrhein, V. et al. (2019) Retire statistical significance. *Nature*, 567, 305-307.

Muff, S. et al. (2022) Rewriting results sections in the language of evidence. *Trends in Ecology & Evolution*, 37, 203-210.

Reviewer #2 (Remarks to the Author)

In this study, the authors explore the combined effect of changes in temperature and nutrient availability on the dynamics of picophytoplankton groups through an experiment using in situ communities in the Atlantic Ocean. The authors selected different zones along the AMT transect with different conditions to investigate the response of these groups to temperature and nutrient variability.

Overall, this work is very interesting and timely, as it addresses a key issue that has many implications in the current climate change situation. The study site selected to conduct the experiment is very convenient to address the question and relevant, as includes one of the most important oligotrophic regions of the ocean, but also a latitudinal transect that includes different in situ conditions.

The experiment is well designed and reported, and the lab analyses are well performed and described. I cannot evaluate nutrient analysis or active chlorophyll a method as I don't have experience with them.

We are very grateful to this reviewer for their constructive and detailed comments. In particular, we agree with the reviewer that the time-scale of the observed responses is an important issue, which is now addressed in the Discussion of the revised manuscript. We have also improved the statistical treatment of the data, following the suggestions from both reviewers. We provide below specific responses to each point raised.

Major concerns

1. The statistical analysis, in my opinion, needs to be revised and/or better reported. It is difficult to evaluate what they have done following the manuscript as it stands. For this reason, some of their statements are not well supported. To be clear, I don't think there is anything wrong with them, but they need to be better reported and, in my opinion the analyses are not sufficient to support some of the statements (see detailed comments below).

We have re-written the relevant passages of the Methods section to clarify the statistical tests conducted (see lines 71-683). We have also added new analyses: a systematic determination of temperature-nutrient interactive effects following the procedure of Piggott et al. (2015) and a two-way ANOVA to assess the interaction effect (Tables S5 and S6).

In addition, the authors should provide more information about the test applied, what type of anova or t.test was used, the variable included and how it was included. What kind of information does the test provide? The threshold used to interpret the tests. I think that it is also crucial to report all the information that the test provides not just the p-value.

As indicated above, the statistical tests conducted are now explained with more detail in the Methods section.

Another way to analyse the data could be to write a model accounting for the hierarchical structure of the data to tease apart the effect of temperature and nutrients on the different response variables accounting for the variability of the different sampling sites. I am not sure how feasible this would be with your dataset, but this could help to understand the effect of temperature on e.g., biomass, the effect of nutrient addition on biomass, and the effect of the interaction between the two factors on biomass accounting for the variability of each station.

The effect of temperature alone and nutrients alone on biomass and fluorescence was assessed with one-way ANOVA, whose results were given in Tables S2 and S3. In the revised manuscript, we have added a two-way ANOVA to assess the interaction between temperature and nutrients (Tables S5 and S6).

2. A concern I have is the time scale of the study, especially with regard to the assessment of the effect of temperature. I think the authors could explain or acknowledge that this study was conducted on a very short time scale and the implications this could have on the results obtained. My experience addressing the temperature dependence of metabolism and growth using different cyanobacteria and phytoplankton species suggests that the species need an acclimation at each treatment before measuring these variables when possible and, even if not possible, the time scale used is several weeks. I am aware that this is not feasible on board. But do you think the lower response of the species to temperature, especially without adding nutrients, could be related to the short duration of the experiment?

We agree with the reviewer that the time scale of the experiments must be taken into account. However, it is not clear that a short duration of experiments necessarily means that the observed effects will be small. Instead, sudden temperature changes and short experimental times can exacerbate negative effects (particularly of warming), because populations may not have enough time to acclimate to the new conditions. In any case, there are multiple examples in the literature of marked phytoplankton growth responses to temperature changes over the time scales considered in our study (1-4 days): Staehr & Birkenland (2006) grew two mesophilic microalgae at three temperatures and found clear differences in growth after only 3 days of incubation (their Fig. 2); Morán et al. (2018) found changes in coastal phytoplankton biomass (relative to the control) already after 2-4 days of incubation under warmer and colder temperature (their Fig. S1); Liu et al. (2020) demonstrated, using 24-h incubations, that the growth of *Synechococcus* increases with temperature, particularly under nutrient-enriched conditions (their Fig. 2b); and Courboules et al. (2021) reported increases in the growth rate of *Picochlorum* and the picoeukaryotes after 2 days of incubation under increased temperature (their Fig. 5).

We therefore conclude that the relatively small temperature effects we observed are not a result of the short incubation time, but reflect the intrinsic low temperature-sensitivity of phytoplankton growth in oligotrophic regions, as reported recently by other studies we cite (Landry et al. 2021, López-Sandoval et al. 2021).

In the revised manuscript, the Discussion now includes a paragraph that addresses the importance of experimental time scales, and the need to conduct longer experiments (lines 440-455):

“The short duration of our experiments may have prevented full acclimation of individuals and populations to changed temperature conditions and thus contributed to the relatively small response of phytoplankton biomass to warming and cooling. Conversely, sudden temperatures changes can exacerbate negative effects on growth, particularly in the case of warming, precisely because thermal acclimation requires longer exposure times (Harvey et al. 2022). In any case, unequivocal phytoplankton growth responses to changes in temperature have been observed over time scales ≤ 4 days both in laboratory cultures (Staehr and Birkenland 2006) and in natural assemblages of coastal regions (Morán et al. 2018, Courboules et al. 2021) as well as the open ocean (Liu et al. 2021). We therefore submit that the comparatively small temperature effects we observed are not due to the short incubation time, but reflect the intrinsic low temperature-sensitivity of phytoplankton growth in oligotrophic regions. This pattern is in agreement with recent experimental and observational evidence indicating limited direct effect of temperature on phytoplankton growth and productivity in oligotrophic waters of the Red Sea (López-Sandoval et al. 2021) and the tropical Indian Ocean (Landry et al. 2021). Nevertheless, additional experiments using longer incubation times are required to fully characterize the responses of tropical phytoplankton to changes in temperature”

We have also modified the first sentence of the Discussion (lines 345-348) to clarify that the general patterns observed correspond to short-term phytoplankton responses:

“Our experiments were conducted over a 3,500-km transect spanning a wide range of oceanographic conditions and yet provided remarkably consistent patterns in the short-term response of phytoplankton to combined changes in temperature and nutrient supply”

Detailed comments

L51. Could you use a more actual reference here?

Flombaum and Martiny (2021) has been added.

L141. Please add a reference / equation for the UNESCO equation.

The relevant reference (Fofonoff and Millard 1983) has been added.

L248. The flow cytometer measures red fluorescence which is a proxy of the chlorophyll content of the cell.

We have added the following sentence (lines 617-618): “The red fluorescence signal was used as a proxy for cellular chlorophyll content”.

L254 is there any reason to remove *Synechococcus* here? Do they overlap?

There can be some overlap in side scatter between *Synechococcus* and the picoeukaryotes, hence using the orange fluorescence signal (which, in the cell size range assessed by the flow cytometric analysis, is specific of phycoerythrin-containing *Synechococcus*) allows to discriminate between the two groups.

L322 Could you please report here the test you used, and p-value obtained?

We have deleted 'significantly' in this sentence. Given that multiple, repeated measurements of physico-chemical and biological properties of the upper ocean were not available (each station was occupied only for a few hours, and only one CTD cast was conducted), the observed differences between stations could not be assessed statistically. However, we did use multiple indicators of trophic status (resource supply index, surface Chl *a* and picophytoplankton biomass concentration, euphotic layer depth), and all of them suggested that the station at 12.7°N was the least oligotrophic.

L354-357 I understand that figure 2 is useful to see general pattern, but I disagree with the authors that the dynamics shown in Fig S3 and S4 are similar in the 4 experiments. The response is quite different between treatments and not all the experiments respond to the treatments at the same time (e.g., Fig S4a). Could you please clarify this statement?

We believe this statement is justified because there are three recurring features in all experiments: i) an early (24-48 h) decrease, ii) a later increase towards the end of the experiments (72-96 h) and iii) a higher final biomass (Chl *a* or picophytoplankton C) in nutrient-enriched treatments compared to the unamended treatments. Thus we consider the dynamics in all experiments to be broadly similar, but certainly not identical. As the reviewer notes, the timing and magnitude of these responses (the initial decrease and the subsequent increase) differs between experiments. Also, as acknowledged in the text, the early decrease was more marked in terms of Chl *a* than picophytoplankton C.

L358 In line with my previous comment. Although figure 2 is useful as an overview, I think it is also good to keep in mind figure S4 and report the exceptions.

In the revised version, we have added a sentence to note that “the initial decrease in picophytoplankton C was particularly strong in the experiment at 28.7°N, and in the experiment at 12.7°N the warmed treatments displayed a much more marked initial decrease than the other treatments” (lines 171-174).

L405-407 Are you referring to temperature treatments without nutrient addition?

This sentence and the one prior to it refer indeed to temperature treatments, without nutrient addition. This is made clear by the fact that the preceding sentence (lines 219-220) states that the effect of warming and cooling was smaller than that of nutrient addition.

What role do you think incubation time plays in this response?

See response below on the role of incubation time.

Figure 4 and 6 Could you change the color coding here? I think it is confusing to use colors associated with temperature here.

As suggested, we have changed the colours in these figures to avoid confusion. The new colours for the cooling and warming treatments are now consistent between Figs. 4 and 6, and Fig. 5.

L447 How did the authors verify this?

t-tests showed that nutrient-enriched treatments had significantly higher F_v/F_m than treatments without added nutrients, both at $t = 72$ h [$t(4) = -4.857, p = 0.008$] and $t = 96$ h [$t(16) = -4.73, p < 0.001$]. These results are now cited in the revised manuscript (lines 265-267).

L516-517 I am not convinced by this statement. First, because the authors do not show any estimation of growth data in the analysis. The authors mainly test the biomass at the end of the experiment.

Although we did not calculate growth rates as such, our data do show the net growth of phytoplankton under each treatment. Because the initial biomass was the same in all treatments, the treatments that showed higher biomass at the end of the experiments were the ones where higher net growth took place. Fig. 3 shows that, irrespective of temperature, nutrient-enriched treatments had significantly higher final biomass than unamended treatments. These differences were assessed with t-tests, as explained in the legend to Fig. 3: “t-test comparisons between nutrient-amended and non-amended treatments for both Chl *a* and picophytoplankton always yielded p values < 0.01 ”.

Second, I am not convinced that the tests presented here can quantify this. They tested whether one treatment was different from another, but not the effect of nutrients and temperature alone.

The effect of temperature and nutrients alone is, in fact, tested when we compare treatments where only temperature has been changed (cooling or warming), or the nutrient-enriched treatment, versus the control. This is done in Figures 2, 3 and 4, which show that temperature had a small and variable effect on phytoplankton final biomass (hence net growth), whereas nutrient addition consistently resulted in increased biomass (Chl *a* and picophytoplankton C) relative to the control (hence increased net growth). Table S2 shows the results of Dunnett’s post-hoc test to assess the difference in biomass between each temperature-nutrient treatment and the control. The p -values in this table indicate that there is much stronger evidence for nutrient effects than for temperature effects.

Third, I believe that incubation time here might play a role in the temperature response of these organisms, especially under nutrient-limited conditions. Therefore, in my opinion, I think the authors should either soften the statement in this regard or acknowledge the limitations of the study.

As indicated above, we have added a paragraph discussing the potential role of incubation time. The literature shows numerous examples of phytoplankton responses to temperature after 2-4 days of incubation, both in temperate, coastal waters and in the subtropical open ocean. We thus submit that the lack of clear and consistent responses to temperature in our experiments, and the fact that nutrient addition did cause clear and consistent responses in terms of Chl *a* and total picophytoplankton C, reflects a fundamental difference between these two drivers in terms of their potential to affect phytoplankton growth in the oligotrophic ocean.

L540-545 I agree with this paragraph, but I suggest including a few sentences about how the response time might be longer in case of temperature change.

The role of incubation time regarding responses to temperature is discussed in the revised manuscript on lines 440-455.

L548 typo temperature

This is not a typo – ‘to temper’ here means ‘to moderate’ or ‘to mitigate’. We have nevertheless replaced it by ‘to moderate’, so as to make the meaning clearer.

L581-582 I am not convinced that the authors may claim this based on the results or at least clarify that this response is in short time scale (less than a week). The authors observe an important increase in biomass when combining temperature and nutrients but also differences between temperature treatments.

This sentence does not state that temperature treatments had no effect, but that the biomass increase (in terms of Chl *a* and picophytoplankton C) in nutrient-enriched treatments took place irrespective of temperature. This can be seen in Figs. 2, 3 and 4, which show multiple instances of similar levels of final biomass in nutrient-enriched treatments that had different temperatures. In the revised manuscript, we clarify that this conclusion refers to a short-term phytoplankton response. The sentence now reads: “A novel finding from our study is that the short-term biomass response of tropical and subtropical phytoplankton assemblages to increased nutrient availability is largely independent of temperature over a 6°C range”.

REVIEWERS' COMMENTS:

Reviewer #1 (Remarks to the Author):

The authors responded well to the reviewers' comments by including additional statistical analyses. This is an important paper in the field that should be published promptly in *Communications Biology*.

- Tatsuhiro Tanioka

Reviewer #2 (Remarks to the Author):

I would like to thank the authors for addressing the reviewers' comments and efforts to improve the manuscript. I think the statistic is better presented now. I also think the authors have done a great job clarifying the results compared to the previous version. I have no major concerns about the article, and I think it will open the door to better explore the temperature dependence of phytoplankton under oligotrophic conditions.

As a minor comment, I would like to point out that most of the work that the authors suggested demonstrates the fitness response of phytoplankton to temperature on a short time scale was done under nutrient-rich conditions. I think we should expect a longer lag phase in the case of oligotrophic environments because species will struggle more to respond to stress than if we "help" them with nutrient enrichment. I think this is worth exploring in the future.